# Multiple Choice Questions and Large Languages Models: A Case Study with Fictional Medical Data

## Abstract

Large Language Models (LLMs) like ChatGPT demonstrate significant potential in the medical field, often evaluated using multiple-choice questions (MCQs) similar to those found on the USMLE. Despite their prevalence in medical education, MCQs have limitations that might be exacerbated when assessing LLMs. To evaluate the effectiveness of MCQs in assessing the performance of LLMs, we developed a fictional medical benchmark focused on a non-existent gland, the Glianorex. This approach allowed us to isolate the knowledge of the LLM from its test-taking abilities. We used GPT-4-Turbo and Claude 3.5 Sonnet to generate two comprehensive textbooks on the Glianorex in both English and French and developed corresponding multiple-choice questions in both languages. We evaluated various open-source, proprietary, and domain-specific LLMs using these questions in a zero-shot setting. The models achieved average scores around 64%, with minor performance differences between larger and smaller models. Performance was slightly higher in English than in French. Fine-tuned medical models showed some improvement over their base versions in English but not in French. The high performance across models suggests that traditional MCQ-based benchmarks may not accurately measure LLMs' clinical knowledge and reasoning abilities, instead highlighting their pattern recognition skills. This study underscores the need for more robust evaluation methods to better assess the true capabilities of LLMs in medical contexts.

## 1 Introduction

Large Language Models (LLMs), such as ChatGPT, have demonstrated significant potential in the medical field, with studies evaluating their performance on tests originally designed for humans, including the USMLE (Jin et al., 2020; Pal et al., 2022; Jin et al., 2019; Nori et al., 2023). Furthermore, domain-specific research shows that these models perform well on specialized medical exams in areas such as pediatrics, radiology, ophthalmology, plastic surgery, and oncology (Rydzewski et al., 2024; Bhayana et al., 2023; Barile et al., 2024; Mihalache et al., 2023; Humar et al., 2023). The common reliance on multiple-choice questions in these assessments reflects their widespread use as a testing method for medical students globally (Al-Wardy, 2010).

However, multiple-choice questions (MCQs), while easy to administer and grade, have notable limitations, often promoting surface learning and pattern recognition over deep understanding (Veloski et al., 1999). Few studies address the potential issues unique to LLMs, such as the reliance on statistical patterns rather than genuine understanding. For instance, Meerkat-7b improved its performance by 18.6% on medical benchmarks through training Mistral 7b on synthetic questions, outperforming Meditron-7b based on Llama2 7b, which saw only a 10.5% improvement despite using a much larger and high-quality dataset of clinical guidelines and articles (Kim et al., 2024; Chen et al., 2023). This discrepancy highlights that extensive multiple-choice question-based training can be more effective than using comprehensive medical content, raising concerns about the true depth of understanding being assessed.

These potential issues are particularly relevant for LLMs, which depend heavily on large datasets that might contain statistical patterns. This dependency can result in models arriving at correct answers

for incorrect reasons, such as identifying skin cancer based on extraneous features like a ruler in the image (Narla et al., 2018). To address these concerns, this study proposes evaluating LLMs using a multiple-choice question test based on entirely fictional medical knowledge. By doing so, we aim to determine whether traditional evaluations are sufficient for assessing the clinical knowledge and reasoning abilities of LLMs for the medical domain, free from the influence of pre-existing data.

## 1.1 RELATED WORK

Evaluating medical knowledge and clinical skills remains an active research area, with new methods such as oral and competency evaluations proposed to better assess medical students and residents (Veloski et al., 1999; Prediger et al., 2020; Goins et al., 2023). Globally, medical evaluations heavily rely on MCQs, like the USMLE in the United States, which significantly influences residency placements (Gauer & Jackson, 2017). LLMs are similarly evaluated using MCQs to assess their medical knowledge. Google introduced MultiMedQA with their Med-PaLM model, combining several existing medical benchmarks, and this has become a standard for evaluating medical proficiency in AI models (Singhal et al., 2023; Pal et al., 2024). Recently, Google incorporated more manual evaluations by medical doctors for their Med-Gemini model (Saab et al., 2024). MultiMedQA includes the following benchmarks:

**MedQA-USMLE**  This subset of the MedQA dataset was sourced from the National Medical Board Examination, the organization responsible for the United States Medical Licensing Examination (NBME, 2024). The dataset is composed of a total of 12723 questions split into a training set of 10178 samples, a validation set of 1273 questions and a test set of 1273 questions. The questions have 4 options with only one correct answer (Jin et al., 2020). Most questions present a clinical vignette and require the test taker to apply clinical or foundational science knowledge to select the best answer.

**MedMCQA**  The Multi-Subject Multi-Choice Dataset for Medical domain is composed of 194k multiple choice questions obtained from the All India Institute Of Medical Science (AIIMS) and National Eligibility cum Entrance Test Postgraduate (NEET PG) entrance exam (AIIMS, 2024; NBEMS, 2024). These questions are split into 3 subsets, one training subset composed of 183k samples, a validation subset of 4.18k samples and a test subset comprising 6.15k samples with the specificity of not containing the correct answer to prevent contamination and manipulation of results. The questions have 4 options each and can be either single or multiple choice. Most questions are straight forward knowledge recall and do not use clinical vignettes.

**PubMedQA**  This biomedical question answering dataset was created using PubMed (NLM, 2024) article abstracts from which the authors derive a question, a context, a long answer and a yes/maybe/no answer. It comprises 3 subsets, one expert-annotated subset of 1k samples, an unlabeled subset of 61.2k samples and an artificially generated subset of 211.3k samples. The generated samples are used to train models, while 500 samples of the expert-annotated subset are used to test the models. This benchmark was designed to evaluate the reasoning capabilities of models when presented with the abstract and a question pertaining to this abstract (Jin et al., 2019).

**MMLU-Medical**  The Massive Multitask Language Understanding dataset contains 57 tasks, of which 6 tasks are used to assess medical knowledge (Clinical knowledge, Medical genetics, Anatomy, Professional Medicine, College Biology and College medicine) (Hendrycks et al., 2021). These tasks were collected by students from publicly available online resources, including USMLE questions as well as undergraduate level questions. The dataset contains 1227 questions split into 25 training samples, 118 validation samples and 1044 test samples. The questions have 4 options with only one correct answer and are a mix of clinical vignettes and recall questions.

## 2 METHODS

We devised a novel approach to assess the relevance of MCQs and LLMs' utilization by generating entirely fictional content. We performed this process twice, once using GPT-4 and a second time using Claude 3.5 Sonnet. In each iteration, we co-created a fictional gland named the Glianorex, located in

the mediastinum and purported to regulate emotions. This ensured no prior knowledge existed about the Glianorex, isolating the models' reasoning capabilities from memorized information.

**Knowledge** For both GPT-4 and Claude 3.5 Sonnet, we began by generating a comprehensive textbook on the Glianorex, detailing its history, physiology, anatomy, and pathology. We structured each textbook using a top-down approach, defining key chapters and subchapters to provide a coherent framework for the respective model. To maintain consistency across the chapters, we generated summaries of key points, such as the anatomical location of the Glianorex and the roles of its hormones, Equilibrion and Neurostabilin, for both textbooks.

**Questions** Based on these fictional textbooks, we used GPT-4 and Claude 3.5 Sonnet separately to generate MCQs. For each model, these questions contained four choices with only one correct answer, adhering to a format similar to that of the USMLE to ensure uniformity. To facilitate the creation of these questions, we designed a prompt for each model that included the table of contents and a paragraph from the respective textbook. See Table 1. In addition, we included a random gender and age between 12 and 90 in 50% of the prompts for both models to ensure variability in clinical vignettes. This approach guided both models to generate questions in a JSON format consistent with existing medical benchmarks. For each model, we used a temperature of 1 and ran 4 generations per paragraph to ensure some variability in questions. After generation, the order of options was randomized to ensure a balanced distribution of correct labels amongst the 4 options.

**Multilingual** To study the influence of language on test taking abilities, we used GPT-4 to translate the generated textbooks and questions using a simple one-shot prompt per paragraph and question asking the model to translate to French.

**Validation** We engaged two medical doctors who took at least one step of the USMLE in the past five years, to assess question quality. They evaluated 100 random English questions on a 7-point Likert scale and answered them to establish a human baseline. We conducted a keyword search for "context" across all questions to identify potential incompleteness. Additionally, a medical doctor manually verified the consistency of Introduction, Anatomy, and Biochemistry chapters in both English and French GPT-4 textbooks to assess both language quality and consistency of the translation.

**Models** To evaluate the performance of LLMs, we selected a diverse set of models, including both proprietary and open-source options. We included commonly used foundational models as reported in Table 2. Additionally, we included two fine-tuned medical domain models based on `mistralai/Mistral-7B-v0.1` to assess the influence of domain-specific training on this fictional benchmark. First, `internistai/base-7b-v0.2` (Apache 2.0) which we trained on a mixture of general data, medical textbooks, and MCQs, demonstrating improved performance on medical evaluations compared to its base model (Griot et al., 2024). Then, `dmis-lab/meerkat-7b-v1.0` (Creative Commons Attribution Non Commercial 4.0), which was trained exclusively on multiple-choice questions, some of which were generated from medical textbooks (Kim et al., 2024). The latter training approach showed a significant performance increase on the benchmarks using a relatively small amount of training data compared to continued pretraining on large datasets of medical data as shown by Meditron and PMC-LLaMA (Chen et al., 2023; Wu et al., 2024).

**Evaluation** The evaluation was conducted using the lm-evaluation-harness in a zero-shot setting, meaning that the models were presented with the questions and choices without any additional training specific to the Glianorex content (Gao et al., 2023). The task was modeled after the MedQA 4 options task, using a log likelihood approach to measure the models' accuracy. The standard error of the mean was then multiplied by 1.96 to obtain the 95% confidence interval assuming a normal distribution of errors. Additionally we assessed the statistical significance of the performance against a random model using a cumulative distribution function on a binomial distribution. Model comparisons were conducted using a two-way analysis of variance (ANOVA), followed by Tukey's honestly significant difference (HSD) test for post-hoc pairwise comparisons of performance. We run these evaluations on a virtual machine with 4 NVIDIA GPU A100 80GB on Microsoft Azure, for a total runtime of 4 hours including model download time.

Table 1: Prompt used to generate multiple choice questions based on a subset of the textbook. The prompt template contains two variables **TABLE OF CONTENT** and **TEXTBOOK PARAGRAPH** which are respectively replaced with the table of content of the textbook and a random paragraph from the textbook to provide context to the model.

| Role | Content |
|---|---|
| System | You are a helpful assistant helping generate knowledge on a fictional gland and its associated diseases. You are tasked with transforming the existing text to generate variations to help learn the content. |
| User | You are given some context and a table of content to help:
**TABLE OF CONTENT**
Query: Generate a very complicated multiple-choice question requiring multiple steps of reasoning with 4 options, these are not reading questions but a test to ensure the student understands and knows the content. Here is an example json output, match this format:

```json
{
  "question": "The question",
  "choices": ["(A) Choice A",
    "(B) Choice B",
    "(C) Choice C",
    "(D) Choice D"],
  "solution": "(D) Choice D"
}
```

Text: **TEXTBOOK PARAGRAPH** |

Table 2: Foundational models included in the study.

| Model | License |
|---|---|
| gpt-3.5-turbo-0125 | Proprietary (OpenAI, 2023) |
| gpt-4-turbo-2024-04-09 | Proprietary (OpenAI, 2023) |
| gpt-4o-2024-05-13 | Proprietary (OpenAI, 2024) |
| 01-ai/Yi-1.5-9B | Apache 2.0 (AI et al., 2024) |
| 01-ai/Yi-1.5-34B | Apache 2.0 (AI et al., 2024) |
| mistralai/Mistral-7B-v0.1 | Apache 2.0 (Mistral, 2024) |
| mistralai/Mixtral-8x7B-v0.1 | Apache 2.0 (Mistral, 2024) |
| meta-llama/Meta-Llama-3-8B | Llama 3 license (AI@Meta, 2024) |
| meta-llama/Meta-Llama-3-70B | Llama 3 license (AI@Meta, 2024) |
| Qwen/Qwen1.5-7B | Tongyi Qianwen license (Bai et al., 2023) |
| Qwen/Qwen1.5-32B | Tongyi Qianwen license (Bai et al., 2023) |
| Qwen/Qwen1.5-110B | Tongyi Qianwen license (Bai et al., 2023) |

By generating entirely fictional content, we ensured that no pre-existing data could influence the models' performance, thus providing a clear evaluation of their reasoning and pattern recognition abilities. This methodology allows us to critically assess whether traditional multiple-choice questions are sufficient for evaluating the the true understanding and clinical reasoning capabilities of LLMs.

## 3 RESULTS

### 3.1 DATASET

The resulting fictional textbooks on the Glianorex were generated using the proposed structure for both GPT-4 and Claude 3.5 Sonnet. Each textbook contains detailed sections on the anatomy, physiology, biochemistry, pathology, and diagnostic tools related to the Glianorex. For both models, the textbooks were produced in English and French, each containing approximately 35,000 words. We then reused paragraphs of the English textbooks to generate multiple-choice questions in English, followed by a translation step to obtain the same questions in French. The GPT-4 process resulted in 264 questions per language, while the Claude 3.5 Sonnet process produced 224 questions per language. For both models, examples of these questions included complex scenarios requiring multiple steps of reasoning. Each question adhered to a four-option format similar to MedQA-USMLE standards, with one correct answer.

Human validation of data quality did not reveal major flaws. Two medical doctors evaluated a sample of 100 English questions, assigning high quality scores (6.94 and 6.86 out of 7), comparable to standard board exams. Manual verification identified only 8 questions (4 per language, ¡1% of total) as incomplete due to missing context. A thorough review of key textbook chapters revealed consistent structure and content across languages, with only minor, inconsequential variations in French abbreviation usage.

### 3.2 EVALUATIONS

All of the models achieved relatively high scores averaging at 63.8%, as illustrated in Figure 1. This score is to put in perspective compared to medical doctors obtaining on average 27% which is within the expected results of answering randomly. A statistically significant difference was noted between the top-performing models and the lowest-performing models as shown in Table 3. The performance differences when isolating languages were also significant and more frequent in English as shown in Tables 6 and 7. We also calculated Cohen's d between all model pairs which revealed a range of effect sizes, indicating varying degrees of performance differences between the models (Cohen, 2013). Most of the comparisons show very small or negligible effect sizes, with many pairs having a Cohen's d close to 0 as shown in Table 8. For instance, pairs such as `01-ai/Yi-1.5-34B` - `01-ai/Yi-1.5-9B` ($d = 0.002$) and `01-ai/Yi-1.5-34B` - `gpt-3.5-turbo-0125` ($d = 0.030$) suggest negligible differences. This pattern is consistent across most pairs, indicating that the models' performances are closely aligned. However, a few pairs demonstrate more noticeable differences, such as `dmis-lab/meerkat-7b-v1.0` - `gpt-4o-2024-05-13` ($d = 0.343$), and `gpt-4-turbo-2024-04-09` - `mistralai/Mistral-7B-v0.1` ($d = 0.254$), suggesting small performance disparities. Overall, the analysis reveals that while some variation exists, the effect sizes for most model comparisons are small. Additionally, the average score for English questions was 65.7%, while for French questions, it was 61.8%. Most models showed better performance in English than in French on this benchmark, with the exception of `meta-llama/Meta-Llama-3-70B`, `gpt-4-turbo-2024-04-09`, and `Qwen/Qwen1.5-32B` achieving similar performances in both languages.

Finetuned models `internistai/base-7b-v0.2` and `dmis-lab/meerkat-7b-v1.0` exhibited improved performance in English compared to their base model, `mistralai/Mistral-7B-v0.1`. However, this improvement was not observed in French, suggesting that domain-specific training enhances performance but the lack of multilingual data in continued training may reduce performance in other languages.

Using a binomial test with a 25% probability of guessing the correct answer, we calculated the probability of obtaining the same score as the lowest performing model and found that the probability was less than $10^{-54}$ . However, the minor performance differences among foundational models of

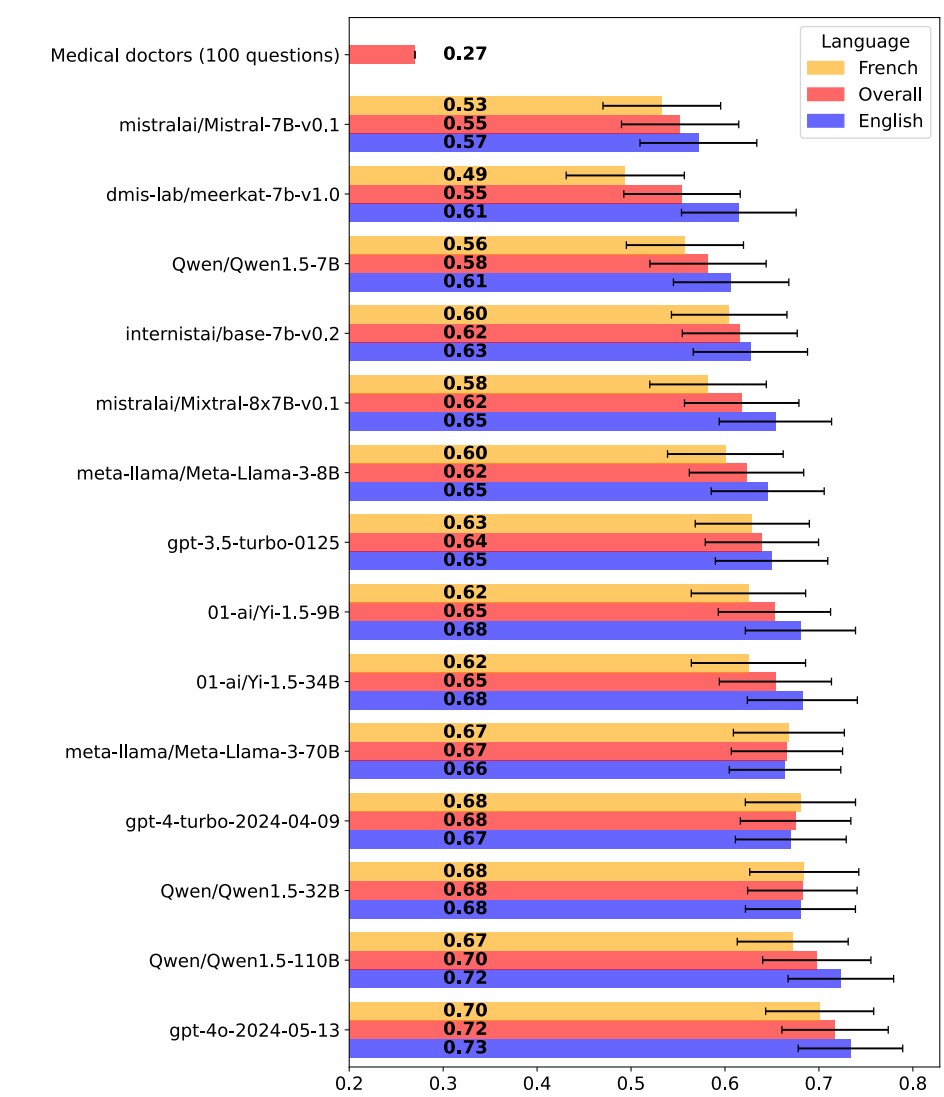

Figure 1: Accuracy of the evaluated models on the synthetic benchmark with a 95% confidence interval. We also display the scores for English and French, highlighting that most models perform better in English than in French.

various sizes and architectures in English indicates that these factors do not substantially affect the ability to answer multiple-choice questions without factual knowledge of the subject.

The distribution of correct answers, as depicted in Figure 2, provides crucial insights into how models handle multiple-choice questions on fictional knowledge. For English, the distribution is heavily skewed towards the top, indicating that a majority of the questions are correctly answered by most models. This suggests that the models, despite their lack of explicit knowledge about the fictional content, are able to leverage their understanding of language patterns and context to select the correct answers.

In contrast, while the distribution for French is still imbalanced, it is less skewed compared to English. This indicates that the models face greater difficulty in applying their inferential and contextual reasoning skills to answer fictional knowledge questions in French. The relatively less imbalanced French distribution highlights that while models can generalize their understanding to some extent, the complexity increases when handling a language other than English.

Table 3: Statistical significance of the performance differences between models (* $p < 0.05$, ** $p < 0.01$, *** $p < 0.001$, and **** $p < 0.0001$).

| | Qwen/Qwen1.5-110B | dmis-lab/meerkat-7b-v1.0 | gpt-4-turbo-2024-04-09 | gpt-4o-2024-05-13 | mistralai/Mistral-7B-v0.1 |
|---|---|---|---|---|---|
| Qwen/Qwen1.5-7B | * | | | ** | |
| meta-llama/Meta-Llama-3-70B | | * | | | * |
| Qwen/Qwen1.5-32B | | ** | | | ** |
| Qwen/Qwen1.5-110B | | ** | | | *** |
| dmis-lab/meerkat-7b-v1.0 | | | * | **** | |
| gpt-4-turbo-2024-04-09 | | | | | * |
| gpt-4o-2024-05-13 | | | | | **** |

## 4 DISCUSSION

The results of this study highlight several key insights into the capabilities and limitations of LLMs in handling MCQs based on fictional medical knowledge. Despite the novelty and complexity of the fictional organ, the Glianorex, and the associated textbook, all evaluated models performed achieved high scores. This finding suggests that LLMs are adept at recognizing patterns and applying test-taking strategies, even in unfamiliar contexts.

**Benchmarking**   The high performance across various foundational models in English, regardless of their architecture, size, or specialization, indicates that traditional MCQ-based benchmarks may not be sufficient for assessing the true understanding and clinical reasoning abilities of LLMs. These benchmarks appear to test the models' ability to identify patterns and associations rather than their genuine comprehension of the material. Consequently, relying solely on MCQs for evaluating LLMs in medical and other specialized domains might lead to an overestimation of their actual capabilities. This finding is additionally supported by research demonstrating that models are becoming less reliable as they are scaled up(Zhou et al., 2024). Using adversarial benchmarks such as the one introduced in this study could help identifying reductions in reliability during development.

**Training**   The improved performance of finetuned models Internist.ai and Meerkat over their base versions underscores the impact of domain-specific training on enhancing LLM capabilities. However, this improvement was predominantly observed in English, which raises questions about the multilingual generalization capabilities of these models.

**Language**   The difference in performance between English and French underscore the models' reliance on language processing capabilities to infer correct answers from multiple-choice options. The less imbalanced French distribution suggests that the models' inferential strategies are less effective when applied to a language they may process less fluently. This variability provides valuable insights into the strengths and limitations of current language models in dealing with fictional knowledge across different languages.

### 4.1 MEDICAL IMPLICATIONS

Current medical evaluation standards may not accurately reflect the capabilities of LLMs in the medical domain, raising significant concerns about their safety and clinical implications in real-world settings. Performance claims based on MCQs could misrepresent the actual capabilities of these models, leading to a false sense of trust that might endanger patients who rely on these systems

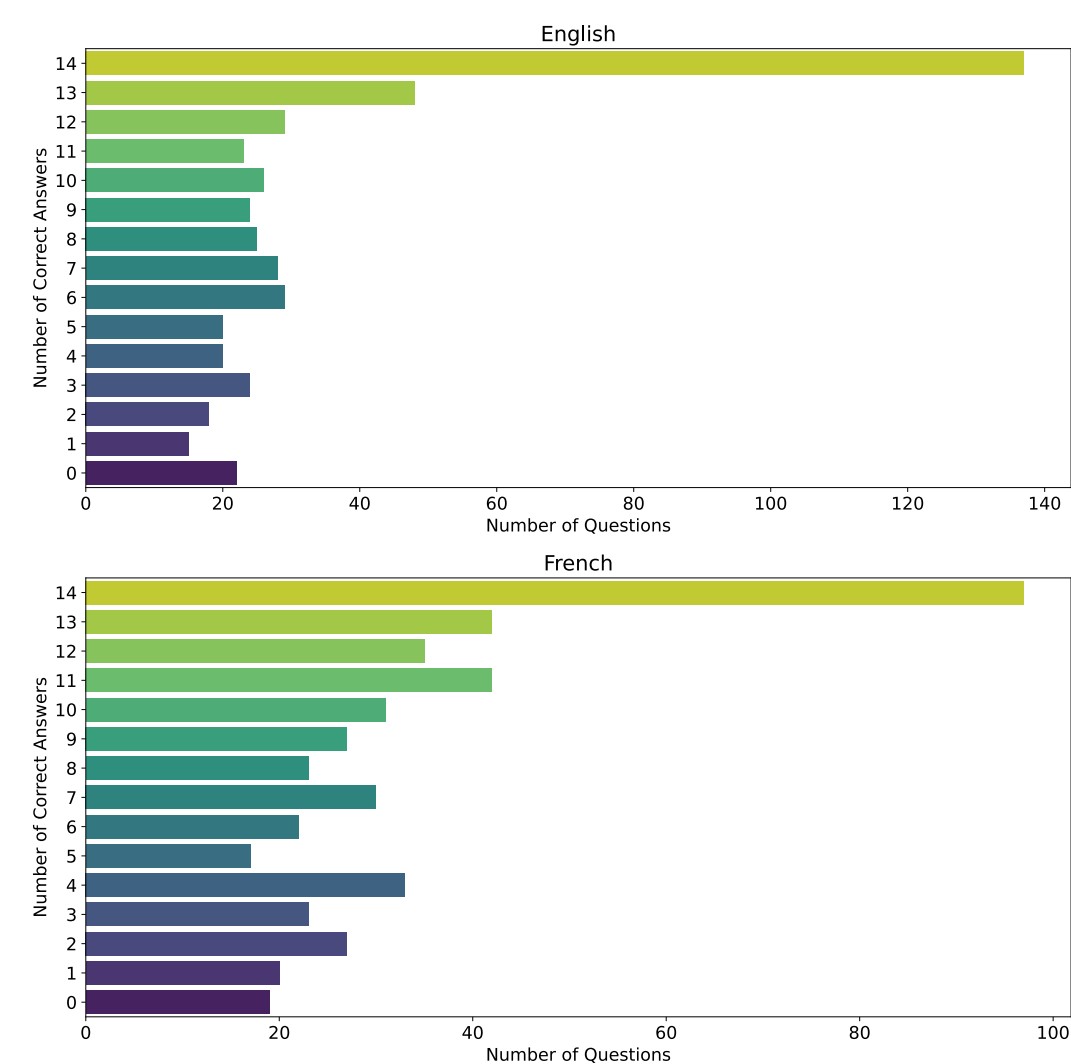

Figure 2: Distribution of correct answers per question. Questions at the top are answered correctly by every model whereas answers at the bottom are answered incorrectly by all models. The distributions are skewed towards the top for both English and French, but the French distribution is more balanced than the English distribution.

instead of consulting their physicians or physicians who implement these systems to provide clinical decision support.

Such claims could also undermine trust within the medical community, which has already expressed skepticism and concerns regarding the application of LLMs in medicine (Marks & Haupt, 2023; Flanagin et al., 2023). Misrepresenting the medical capabilities and usefulness of these models may lead physicians to view artificial intelligence as more of a commercial selling point than a tool for real progress, potentially hindering the adoption of AI and limiting the opportunities for multidisciplinary teams to develop clinically relevant models.

We recommend the inclusion of medical professionals to evaluate the models and for developers to use more caution when making claims based on current benchmarks that may not accurately evaluate medical capabilities. Similarly to medical devices and drugs, models should undergo clinical trials to ensure safety and demonstrate a benefit for patients over current practices (Widner et al., 2023). This requires a change of paradigm and to answer concrete questions such as "Does the use of model X to recommend parenteral nutrition reduce mortality in hospitalized patients with neck cancer?" instead

of the current approach of trying to assess medical capabilities which lacks both a proper definition and relevance for clinical practice.

## 4.2 FUTURE DIRECTION

The findings of this study suggest several avenues for future research. First, there is a need to develop alternative evaluation methods that go beyond multiple-choice questions to assess the deeper understanding and reasoning capabilities of LLMs. These methods could include open-ended questions, scenario-based assessments, and interactive simulations that require models to apply their knowledge in more complex and realistic contexts. While these evaluations are not sufficient to ensure safety and clinical relevance, they would provide a more realistic view of the capabilities on specific domains which could lead to clinical trials.

Second, further exploration of multilingual training and evaluation is necessary to ensure that LLMs can perform consistently across different languages. This is particularly important in the medical field, where accurate comprehension and communication in multiple languages can have significant implications for patient care.

Lastly, investigating the underlying mechanisms that contribute to the enhanced performance of finetuned models could provide valuable insights into effective training strategies. Understanding how additional domain-specific training improves test-taking abilities and clinical reasoning can guide the development of more advanced and capable LLMs.

## 5 LIMITATIONS

**Knowledge coherence** We performed a partial coherence check on the generated textbook to . This oversight could result in inconsistencies or contradictions within the text, potentially creating questions with multiple plausible correct answers depending on the chapter context provided to the model during question generation. However, since the LLMs had no prior exposure to this fictional gland, these potential inconsistencies do not undermine the overall conclusions about the models' performance and does affect the internal validity of the study.

**Sample size** The study generated 488 questions per language, for a total of 976 questions. This sample size is relatively small but is within the same order of magnitude of established multiple-choice benchmarks included in MedQA-USMLE. Despite the limited number of questions, the statistical analysis suggests that the observed performance differences are unlikely to be due to chance.

**Synthetic biases** We used GPT-4 and Claude 3.5 Sonnet to generate the multiple-choice questions, which could introduce hidden patterns that the models might exploit and affect the external validity of this work. To reduce this potential bias, we used two models and a high temperature and generated multiple samples per paragraph. While it is challenging to detect and eliminate all potential patterns, it is important to note that LLMs are trained on extensive datasets and learn a compressed representation of the data which implies that any bias introduced by the models were likely present in the real-world data. Thus, this limitation reflects a realistic scenario where models encounter and utilize inherent patterns in real-world data and does not alter the construct validity of the benchmark. Furthermore, the difficulty experienced by medical doctors in answering these questions suggests that any patterns present are not easily discernible, even to domain experts. This reinforces that the performance of LLMs on this benchmark cannot be attributed to simple, exploitable patterns, as such patterns were not apparent even to highly trained professionals.

**Model selection** Although we evaluated a diverse set of models, including proprietary, open-source, and fine-tuned medical models, the selection was not exhaustive. There may be other models with different architectures or training methodologies that could yield different results. To minimize this bias, we selected models based on their current popularity as being a representative sample of LLMs currently in use. This selection

## 6 CONCLUSION

This study demonstrates that LLMs can achieve high scores on multiple-choice questions based on entirely fictional medical knowledge, even without prior exposure to the content. By using a novel approach of creating a fictional gland, the Glianorex, and generating a comprehensive textbook and related multiple-choice questions, we have isolated the models' reasoning capabilities from their memorization of real-world data. The findings reveal that models of different architectures, sizes, and specializations perform similarly, suggesting that they rely on pattern recognition and test-taking strategies rather than genuine understanding and memorization of the material.

Our results call into question the effectiveness of current multiple-choice question-based benchmarks for evaluating the clinical knowledge and understanding of LLMs. The similar performance across models indicates that traditional multiple-choice questions may not adequately distinguish between superficial pattern matching and deep comprehension. This study highlights the need for developing more robust evaluation methods that better assess the true understanding and reasoning capabilities of LLMs in the medical domain.

In conclusion, while LLMs show promise in handling medical multiple-choice questions, our findings suggest that current benchmarks may not fully capture their clinical knowledge and reasoning abilities. Future research should explore alternative evaluation methods that go beyond current multiple-choice questions to provide a more accurate assessment of Large Language Models' capabilities in medicine and other specialized fields.

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

# A  REPRODUCIBILITY

## A.1  CODE

The additional files attached to this submission contains all the code necessary to reproduce this experiment from scratch. Run instructions are also provided in the README files.

**Synthetic generation**  The code used to generate the synthetic dataset and multiple choice questions is available under the MIT license in the $generator$ directory.

**lm-evaluation-harness**  The main branch of lm-eval-harness contains the $glianorex$, $glianorex\_en$, and $glianorex\_fr$ tasks.

**OpenAI evaluation**  Due to the limitations of lm-evaluation-harness with OpenAI models we had to write OpenAI specific code to evaluate the models available under the MIT license in the $openai - eval$ directory.

## A.2 PARAMETERS

The OpenAI API parameters used to generate the book, translate and generate multiple choice questions are the default parameters as shown in Table 4.

Table 4: OpenAI API parameters

| Parameter | Value |
| --- | --- |
| frequency_penalty | 0 |
| n | 1 |
| presence_penalty | 0 |
| temperature | 1.0 |
| top_p | 1.0 |

## A.3 EVALUATION

To evaluate the open weight models we used lm-evaluation-harness which includes the Glianorex tasks. For any pretrained model hosted on HuggingFace replace $MODEL$ with the path of the model and run the following command:

```
lm_eval --model hf --model_args pretrained=MODEL,dtype="bfloat16",
    parallelize=True --tasks glianorex_en,glianorex_fr --batch_size 32 --
    log_samples --output_path /tmp/results
```

The hardware needed depends on the size of the model, we recommend at least 4 NVIDIA A100 80GB to evaluate models of 70 billion parameters. Reducing the $batch\_size$ can help reduce the memory requirements.

## B ADDITIONAL RESULTS

Table 5: Comparison of model performances depending on language and the model used to generate questions. Bolded values indicate the highest accuracy for the current language. The models compared are gpt-4-turbo-2024-04-09 (GPT) and Claude Sonnet 3.5 (Claude)

| Model | English | | French | |
| --- | --- | --- | --- | --- |
| | GPT | Claude | GPT | Claude |
| 01-ai/Yi-1.5-34B | **0.70** | 0.66 | 0.61 | **0.64** |
| 01-ai/Yi-1.5-9B | **0.69** | 0.67 | **0.62** | 0.62 |
| dmis-lab/meerkat-7b-v1.0 | **0.66** | 0.57 | 0.49 | **0.50** |
| gpt-3.5-turbo-0125 | **0.69** | 0.60 | **0.64** | 0.61 |
| gpt-4-turbo-2024-04-09 | 0.65 | **0.69** | **0.68** | 0.68 |
| gpt-4o-2024-05-13 | **0.74** | 0.72 | 0.69 | **0.71** |
| internistai/base-7b-v0.2 | **0.64** | 0.61 | **0.61** | 0.59 |
| meta-llama/Meta-Llama-3-70B | 0.66 | **0.67** | 0.65 | **0.69** |
| meta-llama/Meta-Llama-3-8B | 0.64 | **0.65** | 0.59 | **0.61** |
| mistralai/Mistral-7B-v0.1 | **0.59** | 0.54 | **0.56** | 0.50 |
| mistralai/Mixtral-8x7B-v0.1 | **0.68** | 0.62 | **0.59** | 0.58 |
| Qwen/Qwen1.5-110B | 0.72 | **0.73** | **0.67** | 0.67 |
| Qwen/Qwen1.5-32B | 0.67 | **0.70** | 0.65 | **0.73** |
| Qwen/Qwen1.5-7B | **0.62** | 0.59 | 0.53 | **0.58** |

Table 6: Statistical significance of the performance differences in English between models (* $p < 0.05$, ** $p < 0.01$, *** $p < 0.001$, and **** $p < 0.0001$).

| | Qwen/Qwen1.5-110B | internistai/base-7b-v0.2 | dmis-lab/meerkat-7b-v1.0 | gpt-4o-2024-05-13 | mistralai/Mistral-7B-v0.1 |
|---|---|---|---|---|---|
| Qwen/Qwen1.5-7B | * | | | ** | |
| meta-llama/Meta-Llama-3-70B | | | | | * |
| 01-ai/Yi-1.5-9B | | | | | * |
| 01-ai/Yi-1.5-34B | | | | | * |
| Qwen/Qwen1.5-32B | | | | | * |
| Qwen/Qwen1.5-110B | | * | * | | ** |
| internistai/base-7b-v0.2 | | | | * | |
| dmis-lab/meerkat-7b-v1.0 | | | | * | |
| gpt-4-turbo-2024-04-09 | | | | | * |
| gpt-4o-2024-05-13 | | | | | *** |

Table 7: Statistical significance of the performance differences in French between models (* $p < 0.05$, ** $p < 0.01$, *** $p < 0.001$, and **** $p < 0.0001$).

| | dmis-lab/meerkat-7b-v1.0 | gpt-4-turbo-2024-04-09 | gpt-4o-2024-05-13 | mistralai/Mistral-7B-v0.1 |
|---|---|---|---|---|
| meta-llama/Meta-Llama-3-70B | * | | | |
| Qwen/Qwen1.5-32B | ** | | | |
| Qwen/Qwen1.5-110B | * | | | |
| dmis-lab/meerkat-7b-v1.0 | | ** | *** | |
| gpt-4o-2024-05-13 | | | | * |

Table 8: Measure of effect size between models using Cohen's d on the overall evaluation (English and French included).

| Model 1 | Model 2 | Cohen's d |
|---|---|---|
| 01-ai/Yi-1.5-34B | 01-ai/Yi-1.5-9B | 0.002 |
| 01-ai/Yi-1.5-34B | Qwen/Qwen1.5-110B | 0.094 |
| 01-ai/Yi-1.5-34B | Qwen/Qwen1.5-32B | 0.061 |
| 01-ai/Yi-1.5-34B | Qwen/Qwen1.5-7B | 0.148 |
| 01-ai/Yi-1.5-34B | dmis-lab/meerkat-7b-v1.0 | 0.204 |
| 01-ai/Yi-1.5-34B | gpt-3.5-turbo-0125 | 0.030 |
| 01-ai/Yi-1.5-34B | gpt-4-turbo-2024-04-09 | 0.046 |
| 01-ai/Yi-1.5-34B | gpt-4o-2024-05-13 | 0.137 |
| 01-ai/Yi-1.5-34B | internistai/base-7b-v0.2 | 0.079 |
| 01-ai/Yi-1.5-34B | meta-llama/Meta-Llama-3-70B | 0.026 |
| 01-ai/Yi-1.5-34B | meta-llama/Meta-Llama-3-8B | 0.064 |
| 01-ai/Yi-1.5-34B | mistralai/Mistral-7B-v0.1 | 0.208 |
| 01-ai/Yi-1.5-34B | mistralai/Mixtral-8x7B-v0.1 | 0.075 |
| 01-ai/Yi-1.5-9B | Qwen/Qwen1.5-110B | 0.096 |
| 01-ai/Yi-1.5-9B | Qwen/Qwen1.5-32B | 0.063 |
| 01-ai/Yi-1.5-9B | Qwen/Qwen1.5-7B | 0.146 |
| 01-ai/Yi-1.5-9B | dmis-lab/meerkat-7b-v1.0 | 0.202 |
| 01-ai/Yi-1.5-9B | gpt-3.5-turbo-0125 | 0.028 |
| 01-ai/Yi-1.5-9B | gpt-4-turbo-2024-04-09 | 0.048 |
| 01-ai/Yi-1.5-9B | gpt-4o-2024-05-13 | 0.139 |
| 01-ai/Yi-1.5-9B | internistai/base-7b-v0.2 | 0.077 |
| 01-ai/Yi-1.5-9B | meta-llama/Meta-Llama-3-70B | 0.028 |
| 01-ai/Yi-1.5-9B | meta-llama/Meta-Llama-3-8B | 0.062 |
| 01-ai/Yi-1.5-9B | mistralai/Mistral-7B-v0.1 | 0.206 |
| 01-ai/Yi-1.5-9B | mistralai/Mixtral-8x7B-v0.1 | 0.072 |
| Qwen/Qwen1.5-110B | Qwen/Qwen1.5-32B | 0.033 |
| Qwen/Qwen1.5-110B | Qwen/Qwen1.5-7B | 0.243 |
| Qwen/Qwen1.5-110B | dmis-lab/meerkat-7b-v1.0 | 0.300 |
| Qwen/Qwen1.5-110B | gpt-3.5-turbo-0125 | 0.124 |
| Qwen/Qwen1.5-110B | gpt-4-turbo-2024-04-09 | 0.049 |
| Qwen/Qwen1.5-110B | gpt-4o-2024-05-13 | 0.043 |
| Qwen/Qwen1.5-110B | internistai/base-7b-v0.2 | 0.173 |
| Qwen/Qwen1.5-110B | meta-llama/Meta-Llama-3-70B | 0.068 |
| Qwen/Qwen1.5-110B | meta-llama/Meta-Llama-3-8B | 0.158 |
| Qwen/Qwen1.5-110B | mistralai/Mistral-7B-v0.1 | 0.304 |
| Qwen/Qwen1.5-110B | mistralai/Mixtral-8x7B-v0.1 | 0.169 |
| Qwen/Qwen1.5-32B | Qwen/Qwen1.5-7B | 0.209 |
| Qwen/Qwen1.5-32B | dmis-lab/meerkat-7b-v1.0 | 0.266 |
| Qwen/Qwen1.5-32B | gpt-3.5-turbo-0125 | 0.091 |
| Qwen/Qwen1.5-32B | gpt-4-turbo-2024-04-09 | 0.015 |
| Qwen/Qwen1.5-32B | gpt-4o-2024-05-13 | 0.076 |
| Qwen/Qwen1.5-32B | internistai/base-7b-v0.2 | 0.140 |
| Qwen/Qwen1.5-32B | meta-llama/Meta-Llama-3-70B | 0.035 |
| Qwen/Qwen1.5-32B | meta-llama/Meta-Llama-3-8B | 0.125 |
| Qwen/Qwen1.5-32B | mistralai/Mistral-7B-v0.1 | 0.270 |
| Qwen/Qwen1.5-32B | mistralai/Mixtral-8x7B-v0.1 | 0.136 |
| Qwen/Qwen1.5-7B | dmis-lab/meerkat-7b-v1.0 | 0.056 |
| Qwen/Qwen1.5-7B | gpt-3.5-turbo-0125 | 0.118 |
| Qwen/Qwen1.5-7B | gpt-4-turbo-2024-04-09 | 0.194 |
| Qwen/Qwen1.5-7B | gpt-4o-2024-05-13 | 0.286 |
| Qwen/Qwen1.5-7B | internistai/base-7b-v0.2 | 0.069 |
| Qwen/Qwen1.5-7B | meta-llama/Meta-Llama-3-70B | 0.174 |

Continued on next page

| Model 1 | Model 2 | Cohen's d |
|---|---|---|
| Qwen/Qwen1.5-7B | meta-llama/Meta-Llama-3-8B | 0.084 |
| Qwen/Qwen1.5-7B | mistralai/Mistral-7B-v0.1 | 0.060 |
| Qwen/Qwen1.5-7B | mistralai/Mixtral-8x7B-v0.1 | 0.073 |
| dmis-lab/meerkat-7b-v1.0 | gpt-3.5-turbo-0125 | 0.174 |
| dmis-lab/meerkat-7b-v1.0 | gpt-4-turbo-2024-04-09 | 0.250 |
| dmis-lab/meerkat-7b-v1.0 | gpt-4o-2024-05-13 | 0.343 |
| dmis-lab/meerkat-7b-v1.0 | internistai/base-7b-v0.2 | 0.125 |
| dmis-lab/meerkat-7b-v1.0 | meta-llama/Meta-Llama-3-70B | 0.230 |
| dmis-lab/meerkat-7b-v1.0 | meta-llama/Meta-Llama-3-8B | 0.140 |
| dmis-lab/meerkat-7b-v1.0 | mistralai/Mistral-7B-v0.1 | 0.004 |
| dmis-lab/meerkat-7b-v1.0 | mistralai/Mixtral-8x7B-v0.1 | 0.129 |
| gpt-3.5-turbo-0125 | gpt-4-turbo-2024-04-09 | 0.076 |
| gpt-3.5-turbo-0125 | gpt-4o-2024-05-13 | 0.167 |
| gpt-3.5-turbo-0125 | internistai/base-7b-v0.2 | 0.049 |
| gpt-3.5-turbo-0125 | meta-llama/Meta-Llama-3-70B | 0.056 |
| gpt-3.5-turbo-0125 | meta-llama/Meta-Llama-3-8B | 0.034 |
| gpt-3.5-turbo-0125 | mistralai/Mistral-7B-v0.1 | 0.178 |
| gpt-3.5-turbo-0125 | mistralai/Mixtral-8x7B-v0.1 | 0.045 |
| gpt-4-turbo-2024-04-09 | gpt-4o-2024-05-13 | 0.091 |
| gpt-4-turbo-2024-04-09 | internistai/base-7b-v0.2 | 0.124 |
| gpt-4-turbo-2024-04-09 | meta-llama/Meta-Llama-3-70B | 0.020 |
| gpt-4-turbo-2024-04-09 | meta-llama/Meta-Llama-3-8B | 0.110 |
| gpt-4-turbo-2024-04-09 | mistralai/Mistral-7B-v0.1 | 0.254 |
| gpt-4-turbo-2024-04-09 | mistralai/Mixtral-8x7B-v0.1 | 0.120 |
| gpt-4o-2024-05-13 | internistai/base-7b-v0.2 | 0.216 |
| gpt-4o-2024-05-13 | meta-llama/Meta-Llama-3-70B | 0.111 |
| gpt-4o-2024-05-13 | meta-llama/Meta-Llama-3-8B | 0.201 |
| gpt-4o-2024-05-13 | mistralai/Mistral-7B-v0.1 | 0.348 |
| gpt-4o-2024-05-13 | mistralai/Mixtral-8x7B-v0.1 | 0.212 |
| internistai/base-7b-v0.2 | meta-llama/Meta-Llama-3-70B | 0.105 |
| internistai/base-7b-v0.2 | meta-llama/Meta-Llama-3-8B | 0.015 |
| internistai/base-7b-v0.2 | mistralai/Mistral-7B-v0.1 | 0.129 |
| internistai/base-7b-v0.2 | mistralai/Mixtral-8x7B-v0.1 | 0.004 |
| meta-llama/Meta-Llama-3-70B | meta-llama/Meta-Llama-3-8B | 0.090 |
| meta-llama/Meta-Llama-3-70B | mistralai/Mistral-7B-v0.1 | 0.235 |
| meta-llama/Meta-Llama-3-70B | mistralai/Mixtral-8x7B-v0.1 | 0.101 |
| meta-llama/Meta-Llama-3-8B | mistralai/Mistral-7B-v0.1 | 0.144 |
| meta-llama/Meta-Llama-3-8B | mistralai/Mixtral-8x7B-v0.1 | 0.011 |
| mistralai/Mistral-7B-v0.1 | mistralai/Mixtral-8x7B-v0.1 | 0.133 |

Table 9: Example of clinical vignette questions in English and French generated by GPT-4 on a random paragraph of the textbook. The correct answer is shown in bold.

| Content |
| --- |
| A 45 year-old male who works night shifts is hospitalized following an episode of severe mood swings and physical tremors. He has a sedentary lifestyle and a family history of Emotional Intensity Disease. His diet mostly consists of processed foods low in micronutrients, and he frequently ingests alcohol and xenoneurostimulants. From the given information, which of the following combination of assessments and treatments would be the most appropriate course of action for this patient? |
| (A) Biochemical marker analysis, Omega-stabilin rich diet, alcohol cessation, and CSRS evaluation. |
| (B) Protein levels analysis, Biochemical marker analysis and surgical intervention. |
| **(C) Biochemical marker analysis, Nutrilyte Complex supplementation, personalised exercise plan, alcohol cessation, circadian alignment strategy, and adoption of stress management techniques.** |
| (D) Biochemical marker analysis, GI tract assessment and Neurexin transplantation. |
| Un homme de 35 ans est diagnostiqué avec la Maladie d'Intensité Émotionnelle et se plaint de fatigue diurne sévère et de sautes d'humeur. Ses enregistrements polysomnographiques montrent des signes d'une architecture du sommeil perturbée, y compris une paralysie du sommeil. Il rapporte une émotivité au réveil et un sommeil non réparateur. Ses échantillons de sérum montrent un niveau élevé de Somnolabilin nocturne et un schéma de sécrétion de Nocturnin perturbé. Compte tenu de ces résultats, quelle méthodologie a probablement été utilisée pour diagnostiquer son état, quelle hormone est probablement associée à sa perturbation du sommeil et à son atonie physique, et quelle pourrait être une stratégie de traitement possible ? |
| (A) Diagnostic avec la Chrono-Enzyme-Linked Immunosorbent Spectroscopy (C-ELIS) d'Elara-Mendoza, l'hormone Nocturnin devrait être associée à ses symptômes et des interventions pharmaceutiques ciblant la synthèse de Nocturnin comme traitement. |
| (B) Diagnostic avec des essais d'électrovalence synaptique, l'hormone Somnolabilin devrait être associée à ses symptômes et des modifications du mode de vie comme traitement. |
| **(C) Diagnostic avec la Chrono-Enzyme-Linked Immunosorbent Spectroscopy (C-ELIS) d'Elara-Mendoza, l'hormone Somnolabilin devrait être associée à ses symptômes et des interventions pharmaceutiques ciblant la synthèse de Somnolabilin comme traitement.** |
| (D) Diagnostic avec des enregistrements polysomnographiques, l'hormone Nocturnin devrait être associée à ses symptômes et la chronothérapie comme traitement. |

Table 10: Example of recall questions in English and French generated by GPT-4 on a random paragraph of the textbook. The correct answer is shown in bold.

| Content |
| --- |
| Considering the detailed anatomy and vascular supply of the Glianorex, which of the following processes best describes how the Glianorex modulates its endocrine functions in response to emotional stimuli? |
| (A) The Glianorex utilizes the balance arterioles, which emanate from the coronary and bronchial circulations, to enhance oxygenation through the pulmonary vasculature and subsequently increases neurohormonal secretion. |
| (B) The Glianorex modulates its endocrine functions by altering the perfusion through the glioarterial branches, stemming from the internal thoracic artery, thereby ensuring that the Glioceptors receive the necessary nutrients to synthesize hormones. |
| (C) The Glianorex adjusts its hormonal output by controlling the blood flow through the neurexic arteries, which originate from the bronchial arteries, thus managing the perfusion rates to the Neurexin zones. |
| **(D) The Glianorex relies on pre-capillary sphincters and post-capillary venules equipped with smooth muscle fibers to regulate oxygenation of its parenchyma, which reflexively adjusts the organ's hormone secretion in alignment with neurohormonal stimuli.** |
| Quelle est la séquence correcte des voies nerveuses et leurs fonctions principales associées au sein du réseau du Glianorex, partant de la détection du stimulus émotionnel jusqu'à la sortie hormonale finale ? |
| (A) Détection via les Gliocepteurs -¿ Intégration par les Globuli Emotoafférents -¿ Traitement par les Ganglions Sentirex -¿ Sortie hormonale avec Equilibron et Neurostabilin |
| **(B) Détection via les Gliocepteurs -¿ Traitement par les Ganglions Sentirex -¿ Sortie hormonale avec Equilibron et Neurostabilin médiée par les Psychoneurexines -¿ Modulation synaptique par le Synaptome Séraphique** |
| (C) Détection via les Globuli Emotoafférents -¿ Traitement par les Ganglions Sentirex -¿ Sortie hormonale avec Equilibron et Neurostabilin médiée par la Voie Gliopathique Primordiale -¿ Modulation de la sensibilité des Gliocepteurs par le Synaptome Séraphique |
| (D) Détection via les Gliocepteurs -¿ Intégration par les Psychoneurexines -¿ Traitement par les Ganglions Sentirex -¿ Sortie hormonale avec le Synaptome et l'Alectorol |

