# OpenReview forum: "Multiple Choice Questions and Large Languages Models: A Case Study with Fictional Medical Data"
_ICLR.cc/2025/Conference — ICLR 2025 Conference Withdrawn Submission_

### Official Review · Reviewer_1MEu · 2024-11-01

**Soundness:** 2
**Presentation:** 3
**Contribution:** 2
**Rating:** 5
**Confidence:** 4

**Summary:**

In this paper the authors generate a multiple-choice QA dataset from a synthetic textbook based on fictional medical knowledge. The authors find that despite performance in humans being at expected chance levels, all language models tested display performance significantly above-chance. This implies that language-model performance on various benchmarks might be the result of general exam-taking ability, rather than testing actual medical knowledge. If true, this could be an important finding to the field of language model evaluations and would warrant further investigation.

**Strengths:**

The authors take a novel and interesting approach to assessing the extent to which language models rely on internal knowledge in benchmark evaluations, including expert evaluation of their synthetic data to ensure the validity of the generated questions. The results may have important implications for the field of language model evaluation and certainly warrant further investigation.

**Weaknesses:**

The results presented here are only for questions in a single domain, related to questions for a synthetic textbook on only a single fictional topic. It is not clear whether these results would generalise to other domains. Additionally, it is possible that the language models used to generate the synthetic text may have imputed this data with real knowledge, which could partially explain the strong performance for every language model tested.

**Questions:**

* I was not entirely clear whether the exam-takers (humans and language models) had access to the textbook prior to answering the questions or was the textbook only used for generating QA pairs. Additionally, did each question provide any context at all?
* It would be useful if the authors could provide more information on the statistical analysis carried out including:
    * How was the standard error of the mean computed? The reported results are for accuracy so what was the mean taken over?
    * Additionally, how was this computed for the human evaluation? The error bars were smaller for the human results which is surprising given that these were based on a smaller sample of questions
    * How was the 2-way ANOVA carried out? What were the 2 factors used?
* The authors provide some proxy metrics for alignment between model’s answers, such as performance differences, however why did the authors not test answer alignment more directly, using a metric such as Cohen’s kappa? This would allow significantly more insight into how similar performance was between models.
* I’m not sure how informative figure 2 is. If performance of most models was >0.6 accuracy it’s somewhat given that the correct answers would be skewed towards a majority correct. Using metrics such as Cohen’s kappa or a similar approach would be more informative to understanding model agreement. Additionally, this figure would be easier to understand if the x axes were the same scale.
* Why did the authors provide no comparison of human answers with model answers? Given that they only used 100 questions as their sample, it’s possible the sample used to test human performance was a harder subset of questions.
* It would be helpful to see more in-depth analysis of the QA pairs the models got correct and incorrect. For example, were correct answers more likely for specific sub-topics? Were questions based on particular textbook sections more likely to be answered correctly?

---

### Official Review · Reviewer_7Mkn · 2024-11-02

**Soundness:** 3
**Presentation:** 2
**Contribution:** 2
**Rating:** 5
**Confidence:** 4

**Summary:**

This paper addresses an key problem: the inadequacy of current multiple-choice question-answering evaluation methods for Large Language Models (LLMs) in medical contexts. It highlights that LLMs' test-taking abilities, i.e., their capability to rely on statistical patterns rather than understanding, impact their performance on medical benchmarks. This phenomenon is particularly relevant in settings where LLMs may select correct answers based on learned statistical associations rather than true comprehension. To demonstrate this, the authors devised a test based on fictional medical knowledge, effectively removing the influence of prior knowledge for the models. The results indicate that LLMs still perform well on questions about fictional medical knowledge. The authors claim that their finding suggest that LLMs have good test-taking ability that prevent current evaluation methods from effectively assessing the true clinical reasoning capabilities of these models.

**Strengths:**

1. The introduction of a fictional gland is a novel way to control for LLMs’ memorized real-world knowledge. The knowledge about the fictional gland is generated in a comprehensive and detailed way.

2. Human experts were involved to ensure the quality of the generated medical knowledge and questions, and to provide a human performance baseline. Expert involvement makes the experiments more accountable and trustworthy. This is important for applications in healthcare and medicine because these are mission-critical scenarios.

3. The study uses statistical analysis to provide solid evidence supporting its claims.

4. The inadequacy of MCQ evaluation method is a long-standing and important issue, especially in clinical scenarios. The investigation is well-motivated. In addition, the authors clearly outline the motivation for the study and the implications of their findings, making it easy for readers to grasp the significance of the research.

**Weaknesses:**

1. The claim that "entirely fictional content" removes pre-existing data influence is overstated. While the fictional setup minimizes real-world knowledge's impact, it does not fully eliminate it. The generated content of the fictional gland using LLMs stills adhere to the knowledge framework of medicine. Thus, there is pre-existing foundational medical knowledge from LLMs embedded in the content and questions even though no pre-existing data specific for the fictional gland exists. LLMs’ reasoning ability with prior knowledge is not entirely ruled out. This claim, central to the paper’s message, is thus an overstatement. This is a limitation of the method in isolating LLMs' test-taking abilities.

2. The statistical analysis is appreciated, but the authors do not clearly explain the apparent disparity between the small and negligible performance differences derived statistically and the visible differences shown in Figure 1. This may confuse readers unfamiliar with advanced statistical analysis.

3. A known issue with multiple-choice evaluation is selection bias/position bias for option order and IDs. Studies show LLMs may have positional preferences (e.g., favoring option A), making it necessary to control for option order, as positional bias could affect model performance significantly. The authors do not address this issue.

4. The discussion on the distribution of correct answers and models' "inferential strategies" lacks clarity. Terms like "inferential strategies" are uncommon in this field and require further elaboration. Additionally, the paragraph addressing language in the discussion section could benefit from better clarity.

5. While the paper highlights issues with current multiple-choice evaluation, it does not propose and develop solutions. A method that can mitigate the issue and suggest potential paths forward would make the contribution more impactful, moving beyond identifying a problem to offering ways to address it.

6. Presentation flaws are present, such as incomplete sentences in the limitations section, which disrupt readability. For instance, the "Knowledge coherence" paragraph in the limitations section is unfinished, and the "Model selection" paragraph in the same section ends abruptly with "This selection."

**Questions:**

1. Is there existing work on benchmarks for specific medical domains, such as oncology? Including one in the related work section could enhance context.

2. In the discussion's training paragraph, you mention “the improved performance of finetuned models Internist.ai and Meerkat over their base versions underscores the impact of domain-specific training on enhancing LLM capabilities.” Could you clarify what you mean by "LLM capabilities" in this context?

3. Is the generated textbook open-sourced? If so, please provide a link to the anonymous version of the textbook.

---

### Official Review · Reviewer_Qaz8 · 2024-11-03

**Soundness:** 2
**Presentation:** 2
**Contribution:** 2
**Rating:** 3
**Confidence:** 4

**Summary:**

This paper presents a study examining the effectiveness of multiple-choice questions (MCQs) in evaluating Large Language Models' (LLMs) medical knowledge and reasoning capabilities. The authors devise an experiment where they use GPT-4 and Claude 3.5 Sonnet to generate textbooks about a fictional gland called "Glianorex," along with corresponding multiple-choice questions in both English and French. They then evaluate various LLMs, including proprietary, open-source, and domain-specific models, on these questions in a zero-shot setting. The models achieve surprisingly high average scores of around 64%, despite having no prior knowledge of this fictional medical content.

This points to a significant methodological concern: since both the content and questions were generated by LLMs, the high performance likely demonstrates LLMs' ability to recognize and reconstruct patterns in LLM-generated text rather than any genuine medical reasoning capabilities. While the authors interpret their results as evidence that MCQ-based evaluations may not adequately assess medical knowledge, their experimental design inadvertently reveals more about LLM-to-LLM pattern recognition than about the limitations of multiple-choice testing in medical AI evaluation. This highlights a broader issue in AI evaluation methodology, where the tools used to test AI systems may be inherently biased towards the systems' pattern-matching capabilities rather than their actual understanding or reasoning abilities.

**Strengths:**

The study evaluates a comprehensive range of models - from proprietary systems like GPT-4 to open-source models like Mistral and domain-specific medical models. The multilingual evaluation in both English and French adds valuable cross-linguistic insight. Testing across model scales (from 7B to 110B parameters) and comparing both base and medically fine-tuned versions provides a thorough performance landscape.
The core concept of testing reasoning gaps through fictional medical content is important given the high-stakes nature of healthcare applications. With the increasing deployment of LLMs in medical settings, understanding their limitations in medical reasoning versus pattern matching becomes critical for patient safety.
The experimental setup allowed for controlled model performance testing without the confounding variable of pre-existing medical knowledge. The methodological approach of creating a fictional medical domain to isolate reasoning capabilities represents creative thinking about AI evaluation challenges.
The statistical analysis is solid, including Cohen's d effect sizes and significance testing, provides some quantitative backing for the findings.
Code is openly available and use of harness is good practice.
The authors are also transparent about the limitations of their approach.

**Weaknesses:**

1. LLM Chain of Generation:
The experimental pipeline (textbook → questions → answers) is LLM-generated, creating a closed loop that primarily tests LLM-to-LLM pattern recognition. This begs the question if this tests more than other approaches that have shown models fragility to lexical substitution or paraphrasing methods.

2. Well-Established Lack of Reasoning:
Multiple studies have already definitively shown LLMs don't perform actual reasoning:
- GSM8K and other math reasoning benchmarks show LLMs struggle with novel mathematical reasoning
- "Physics of Language Models" work shows LLMs operate through pattern matching rather than causal understanding
This study doesn't meaningfully advance beyond these existing findings.

3. Missing Comparisons and related work:
- No comparison between performance on these fictional medical questions versus real medical benchmarks (like USMLE)
- No comparison to human-generated alternative versions e.g. drug name substitution benchmarks
- Related work should include work on lexical substitution https://aclanthology.org/D14-1066.pdf , dataset contamination https://arxiv.org/abs/2404.18824 and robustness e.g. https://arxiv.org/abs/2406.06573, https://arxiv.org/abs/2406.12066

4. Missed Opportunity for Medical Evaluation:
Instead of showing that LLMs don't reason (which we know), they could have investigated what specific patterns in medical MCQs make them vulnerable to exploitation by LLMs, which would be more useful for improving medical testing.

**Questions:**

LLM Generation Chain:
- How do you control for the fact that both content and questions are LLM-generated?
- Have you considered comparing performance between LLM-generated questions and human expert-generated questions about the same fictional content?

Relation to Known Limitations:
- How does this work advance our understanding beyond existing studies showing LLMs lack reasoning capabilities (GSM8K, Physics of Language Models, etc.)?
- Could you compare your findings about medical MCQ performance to similar pattern-matching behaviors documented in math, physics, and symbolic reasoning tasks?
- What makes medical MCQs different from other domains where LLM pattern-matching has been studied?

Benchmark Comparisons:
- How does model performance on your fictional medical questions compare to performance on real medical benchmarks like USMLE?
- Could you analyze if models exploit similar patterns in both real and fictional medical questions? e.g. Have you considered creating paired real/fictional questions with matched reasoning requirements to isolate the effect of domain knowledge vs. pattern matching?

---

### Official Review · Reviewer_pkgk · 2024-11-04

**Soundness:** 2
**Presentation:** 2
**Contribution:** 1
**Rating:** 3
**Confidence:** 5

**Summary:**

This manuscript aims to assess the QA performance of LLMs on 'fictional' data. Authors generate a textbook about an inexistent gland, generate QA from paragraph fragments from this generated textbook and subsequently evaluate the QA using LLMs in a zero-shot setting.

**Strengths:**

- The study generates a 'fictional' textbook on a fictional gland and uses the synthetic data to evaluate model performance on QA.
- I found it fascinating that the doctors found no major flaws in the generated QA.
- The study evaluates a variety of LLMs (in zero-shot setting).

**Weaknesses:**

- It is known that evaluations for using LLMs for healthcare are insufficient and incomplete.  Newer, better ways of benchmarking them are necessary, for e.g., using real life patient data. LLMs, due to their size and scale are known to memorize their training data. LLMs performing well on QA benchmarks indicate they encode the knowledge (complete/incomplete). With this case study, the aforementioned evidence is only revalidated and makes a limited case for novelty. As mentioned in the future work section, may be interactive, scenario-based assessments are an interesting direction!

- Line 216- 'By generating entirely fictional content, we ensured that no pre-existing data could influence the models’ - this seems hard to be convinced of- even if the gland is fictional, the model is exposed to data about the other organs/biological aspects of the human anatomy that play a role in relation to this gland. The LLM may also be exposed to information/vocabulary on how a gland is supposed to function and what their possible medical conditions that occur to other glands. This could also be a possible explanation as to why models show better performance than a random baseline.

- Line 488 in conclusion: 'This study demonstrates that LLMs can achieve high scores on multiple-choice questions based on entirely fictional medical knowledge, even without prior exposure to the content.' I disagree that this knowledge is completely fictional. Although the gland is fictional, the information of how 'a gland' is supposed to functional is not fictional medical knowledge along with other information about the human anatomy.

- Yes, the models was able to do well on ~80-90 questions. What about others? Could the authors discuss errors? Where did the models go wrong? Were the errors the similar to the ones highlighted in previous work in MedQA? Why were they hard? Did the medical experts find them easier?

- I find the quality of the generated QA may not be completely equivalent/ comparable to existing work-
 Minor corrections
- Line 457: missing text
- Line 485: incomplete sentence

**Questions:**

- In Line 55-58 in the introduction- 'To address these concerns, this study proposes evaluating LLMs using a multiple-choice question test based on entirely fictional medical knowledge. By doing so, we aim to determine whether traditional evaluations are sufficient for assessing the clinical knowledge and reasoning abilities of LLMs for the medical domain, free from the influence of pre-existing data.' This seems to indicate the motivation of the study. Can the authors say more? What were the hypotheses they wanted to study? Did they expect the model to perform poorly? and why? By creating a similar QA study on are they establishing anything different?

- Is it possible to provide more information on the development of the 'Glianorex' textbook? How were the models steered to generate the content?

- There is very limited information on how the models are set up experimentally- i.e. what was the input uniform across all 14 models given the differences in context windows?

- How long were the paragraphs used for generating the questions/options? Is there a chance that not all the questions that were generated were about the fictional gland?

- Can the authors comment more on why the generated questions were difficult for medical professionals? Is it because they were unfamiliar with the working of the fictional gland? or because the answers were linguistically hard to disambiguate? Can they provide examples?

---

### Note · Authors · 2024-11-18

**Comment:**

We want to thank the reviewers and chairs for their feedback. Considering the reviews, we believe our work may not align with the audience of this conference and plan to resubmit it to a more medically focused venue.

**Withdrawal Confirmation:**

I have read and agree with the venue's withdrawal policy on behalf of myself and my co-authors.